# Automated 3D Reconstruction Using Optimized View-Planning Algorithms for Iterative Development of Structure-from-Motion Models

**Samuel Arce** [1] , **Cory A. Vernon** [1] , **Joshua Hammond** [1] , **Valerie Newell** [1] , **Joseph Janson** [1] , **Kevin W. Franke** [2] and **John D. Hedengren** [1,*]

[1] Department of Chemical Engineering, Ira A. Fulton College of Engineering and Technology, Brigham Young University, 350 Clyde Building, Provo, UT 84602, USA; samuelarce@byu.edu (S.A.); cvernon3@byu.edu (C.A.V.); joshua.hammond@byu.edu (J.H.); ven22@byu.edu (V.N.); jhjanson@byu.edu (J.J.)

[2] Department of Civil and Environmental Engineering, Ira A. Fulton College of Engineering and Technology, Brigham Young University, 368 Clyde Building, Provo, UT 84602, USA; kevin_franke@byu.edu

[*] Correspondence: john.hedengren@byu.edu

**Abstract:** Unsupervised machine learning algorithms (clustering, genetic, and principal component analysis) automate Unmanned Aerial Vehicle (UAV) missions as well as the creation and refinement of iterative 3D photogrammetric models with a next best view (NBV) approach. The novel approach uses Structure-from-Motion (SfM) to achieve convergence to a specified orthomosaic resolution by identifying edges in the point cloud and planning cameras that "view" the holes identified by edges without requiring an initial model. This iterative UAV photogrammetric method successfully runs in various Microsoft AirSim environments. Simulated ground sampling distance (GSD) of models reaches as low as 3.4 cm per pixel, and generally, successive iterations improve resolution. Besides analogous application in simulated environments, a field study of a retired municipal water tank illustrates the practical application and advantages of automated UAV iterative inspection of infrastructure using 63% fewer photographs than a comparable manual flight with analogous density point clouds obtaining a GSD of less than 3 cm per pixel. Each iteration qualitatively increases resolution according to a logarithmic regression, reduces holes in models, and adds details to model edges.

**Keywords:** Structure-from-Motion; Unmanned Aerial Vehicles; iterative inspection; automated inspection; multi-scale; view-planning; unsupervised machine learning; autonomous flight; iterative optimization

## 1. Introduction

Recent advancements in remote sensing technologies motivate the use of Unmanned Aerial Vehicles (UAVs) in a variety of aerial imaging tasks [1–5]. One remarkable use of UAVs as remote sensors is the high resolution topographic models through means of inexpensive equipment and a photogrammetric technique known as Structure-from-Motion (SfM) [6]. SfM organizes the spatial information of multiple images, and structures a 3D model from the various surfaces based on relative distances between common points [7]. The success of SfM and UAVs heightens the interest in developing robust frameworks for industrial monitoring and inspection. Researchers have addressed tasks such as performance in unknown environments [8], computational requirements [9], autonomy [10], plant breeding [11], deep learning in agriculture [12], multi-spectral imaging [13], and view planning [14,15].

This study introduces an iterative modeling strategy that provides UAVs with additional autonomy in photogrammetry missions. First, an initial incomplete model is generated with a minimal amount of photos. The photos are processed, and the resulting incomplete point cloud is analyzed to find deficiencies in the model and calculate an optimal set of views. These subsequent views improve model resolution in each iteration of UAV missions. The photos from each iteration combine with previous models to render a more complete point cloud. The analysis repeats until the model achieves a predefined orthomosaic resolution.

### 1.1. Iterative Modeling

Next best view (NBV) algorithms in unknown environments and iterative modeling enhance the pragmatism and quality of UAV-based SfM. Iterative modeling begins with an initial UAV mission that creates a rough model to plan smarter successive flights for data collection and construction.

Schmid et al. utilize a low resolution digital elevation model (DEM) for an initial flight and create additional models from the acquired data [16]. While Schmid et al.'s work solidifies the concept of iterative modeling, additional work quantifies and expands the basic concepts. UAVs do not only fly over long linear features in Martin et al. but also target specific objects and features of interest while not sacrificing the requirement of overall coverage of the environment [17]. Hoppe et al. exploit *a priori* knowledge of environments to achieve high quality models with fewer inputs per flight [18]. Better utilization of *a priori* understanding emerges from another work by Martin et al. as a genetic algorithm applied to an existing DEM achieves 143% of a comparable nadir grid flight [14]. However, returning to Schmid et al., a key limitation to improved SfM through UAVs is the hardware requirements [16]. Fortunately, continued advances in computing and technology give greater capacity to hardware at lighter UAV payloads. The possibilities for future UAV photogrammetry and iterative models will continue to grow from ingenuity and access to more powerful tools.

Another issue addressed by iterative modeling is enhanced capability to capture the essence of complex 3D objects or features that are sharp or obscured (e.g., the undersides of bridges and other overhangs are difficult to reconstruct with automated UAV photogrammetry). Two such examples on bridge case studies are Morgenthal et al. and Pan et al. [19,20]. In these studies 3D point clouds serve as DEMs and assist in eventual model analysis. Another example, though not truly a complex 3D object nor an overhang–is a multi-scale model of the Tibble Fork Dam in Utah County, UT, made possible by the DEM of the scene whose sub-sampled point cloud lends the confidence to safely fly the UAV on successive flights closer to the dam, model quality at key points of interest and maintain the integrity of the entire model without inordinate amounts of data input or time [15].

Iterative modeling reduces probabilistic planning, but much of the previous research delves into coping with uncertainty in unfamiliar environments. As early as 2000, Roy et al. notes that a 3D object is difficult to fully capture from just one view and explores the concept of NBV [21]. By analogy, it is difficult to fully capture a 3D environment from just an initial unrefined model that lacks iterative refinement. Trummer et al. later continued NBV approaches with remote sensing hardware and sought to minimize absolute error in models [22]. Modular optimization addresses uncertainty for autonomous underwater vehicles (AUV) by Hollinger et al. and parallels UAV applications by extension [23]. Meng et al. approached the problem with multiple stages of NBV and SfM [24]; then Faigl et al. sought to optimize multiple 3D goals (for the traveling salesman problem (TSP), which is frequently included as part of UAV path optimization in photogrammety)–new methods to iterate boast the capacity to get similar but faster results with fewer inputs than non-iterative approaches [25]. However, the above approaches trend towards unnecessary complexity if a reasonable *a priori* 3D model of a site remains available. Even a first iteration of a simple nadir flight can recreate most landscapes to moderate resolutions.

Using an existing DEM permits UAVs to safely obtain higher quality views for subsequent optimal reconstructions. Iterative modeling further advances the starting point for NBV and other approaches.

### 1.2. Optimized View Planning

Historically, *a priori* data and models serve as the basis for UAV view and flight planning. In Palmer et al., UAV view planning substitutes the traditional Light Detection and Ranging (LiDAR) approach [26]. A few works utilize targeted 3D modeling, object recognition, and genetic algorithms to pragmatically advance view planning [14,17,27]. As new technologies continue to develop and applications adjust towards optimal view planning [28], NBV applied to *a priori* models or even an NBV approach without prior data to construct high resolution models is a logical step forward for UAV SfM.

SfM requires views from multiple angles of the same feature so that the location can be triangulated from the cameras. A nadir pattern grid flight overlaps photographs at a uniform downward angle to triangulate locations. Obtaining sufficient overlap can be difficult using the nadir method. Oblique camera views can identify more points through feature detection, and with varied angles of view, can produce a more dense and accurate point cloud–note commercial/industrial oblique angles (as with DJI Terra: https://www.dji.com/dji-terra/info) acquire views and coverage that a nadir grid cannot. Oblique views allow for fewer viewpoints with comparable to better coverage than less optimal methods.

In addition to coverage, computational feasibility requires minimizing the amount of photos (viewpoints) taken; viz Nikolic et al., as the number of photos $n$ doubles, the computational resources necessary to create the model increase on the order of $n^2$ [2]. Hausamann et al. show that the number of some photos can be removed from the model construction process while maintaining model quality, but the computational demands are still on the order of $n^2$ for $n$ photos taken [4]. Okeson shows that an optimal set of camera locations can contain the necessary coverage to construct a model using SfM without unnecessary photos [29]. Okeson used existing knowledge of the terrain to create a set of possible camera locations [29]. Then analyzing necessary overlap, the distinct angles necessary for SfM recreation, and previous cameras already selected, chose the next best camera location and added it to the optimal set. The method conducted by Okeson, seeks 95% coverage of the known model through the view selection [29].

It is important to distinguish that *a priori* knowledge of the region modeled is necessary for optimization through Okeson's method [29]. In the current work, limited knowledge of the area in question is assumed or necessary. Instead, a novel approach for visualization and identification of necessary cameras, a NBV approach to view planning that leverages machine learning, is explored.

Machine learning seeks to organize and understand vast quantities of data as described by Jordan M. I. [30]. A computer system is shown many examples and "learns" how to obtain the best answer through pattern recognition and curve fitting [30]. Section 2: *Methods* further elaborates the specific aspects of machine learning for an iterative NBV approach. Through machine learning, the software "learns" from experience and develops paths to achieve desired outputs. The "learning" consists of combining genetic, clustering, and principal component analysis algorithms to refine UAV photogrammetry with continued iterations. This allows the computer to organize the work instead of the user manually inputting each test to improve a model.

While outside the scope of this work, additional issues of the camera network problem and accuracy could be explored in future research. Work by Grün notes how self-calibration of images from SfM leads to model errors [31], and camera network geometry could be considered but methods to ensure accuracy exist as explained in Nocerino et al. [32]. James and Robson describe "vertical doming" during SfM model construction as a result of parallel images self-calibrating erroneously; however, the images in this study avoid this issue by choosing camera views that take advantage of oblique angles [33].

### 1.3. Novel Contributions

The following novel contributions are included in this work:

- An iterative strategy to UAV-based SfM without the use of previous models to initialize the UAV mission
- Demonstrated automated iterative mapping of different terrains with convergence to a specified orthomosaic resolution in simulated environments
- A field study that illustrates the advantages of iterative inspection of infrastructure using UAVs.

## 2. Methods

### 2.1. Iterative Modeling Overview

The authors present a novel approach for mapping different terrains, including industrial equipment, for which no *a priori* models or DEMs are available or existing DEMs present deficiencies. For the purposes of this study, deficiencies in the model are defined as areas where the model is incomplete (i.e., holes in the model). The algorithm uses UAVs and photogrammetry in an iterative fashion, the approach optimizes the next set of best views that will improve the model's orthomosaic resolution. Figure 1 represents the general workflow of the iterative modeling algorithm.

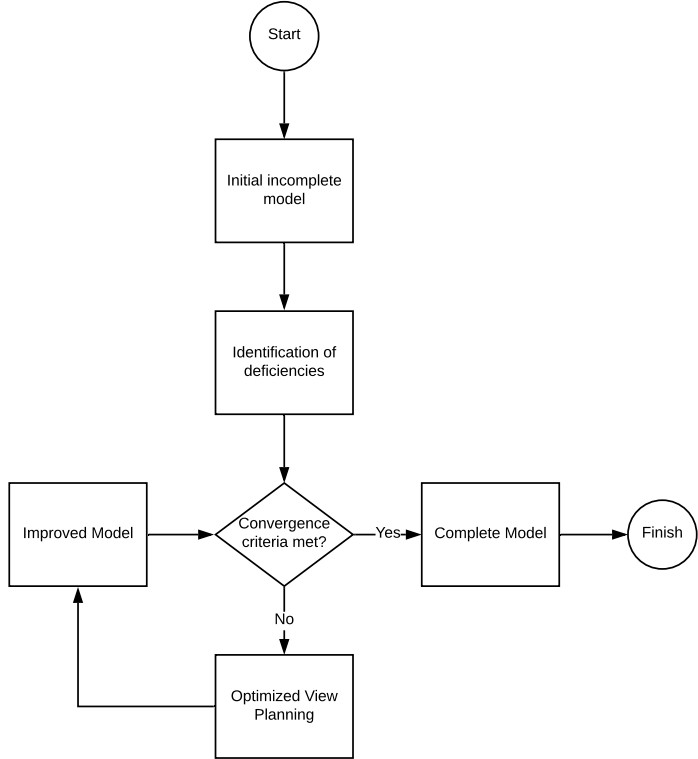

**Figure 1.** General workflow for proposed iterative modeling strategy.

### 2.2. Selection of the Area to Be Mapped

The algorithm initializes by incorporating a set of pictures with enough overlap to render an initial dense cloud. The initial set of photos should be as small as possible to expedite the rendition of the point cloud and is collected using a grid pattern that will cover the area of interest with 85% overlap between photos. Figure 2 shows an incomplete model, as apparent from the numerous "holes" in the model. The model originated from a nadir flight in a simulated wilderness environment.

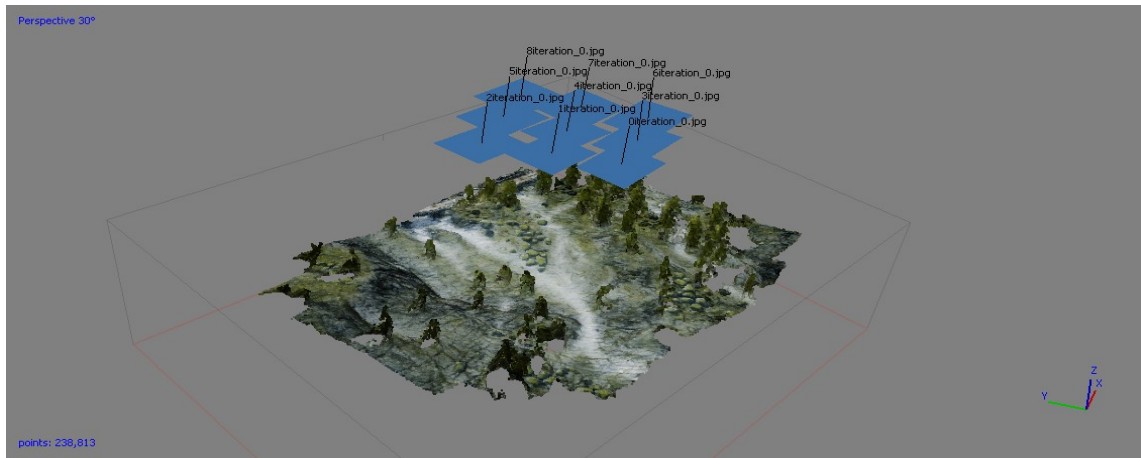

**Figure 2.** Initial model using 9 photos. The position and altitude of the photos are calculated for 85% overlapping.

## 2.3. Identification of Model Deficiencies

After the initial dense cloud has been produced and downsampled to 20,000 points, the deficiencies in the model are prelabeled by identifying boundary points. There has been extensive research on the identification of boundary points in a three dimensional space, such as the *quickhull* method [34], the *alpha-shape* method [35], kernel regression [36], $k$ nearest neighbors (*k-NN*), and principal component analysis (PCA) [37], and the boundary point detection algorithm proposed by Mineo et al. that incorporates *k-NN* analysis and geometric constraints [38]. This study uses a *k-NN* algorithm to find boundary points. Boundary points are identified by their distance to the centroid of the neighborhood, which must be larger than a specified threshold to qualify as a boundary point. The extra level of complexity offered by other boundary point algorithms has a significant impact in other applications such as segmentation and classification, where the distinctiveness of geometric features plays a central role [39]. In our study, the identification of a boundary point approximates where the previous iteration failed to gather enough data for a complete model.

Figure 3 shows the calculated boundary points for an iterative mission in a simulated environment. Note that the edges of the model are identified as boundary points. In many applications, it is desired to constrain the calculation of camera poses to a specific area. In this experiment, only the boundary points that fall within 80% of the area covered by the initial set of pictures are considered.

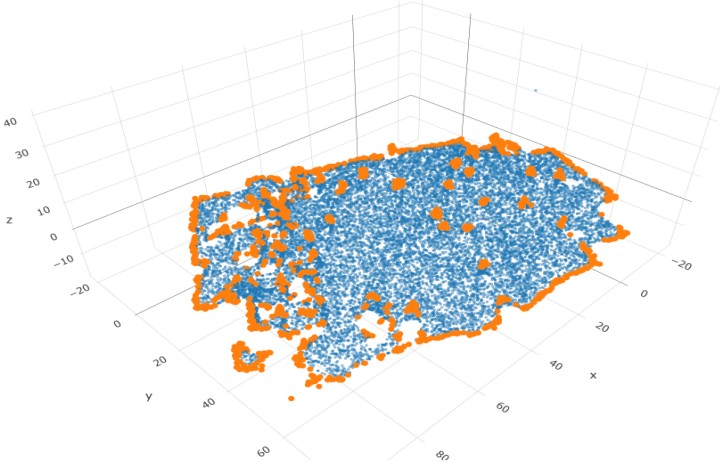

**Figure 3.** Boundary points identified by $k$ nearest neighbors (*k-NN*).

Experiments in simulated environments show that values in the range of $k = 30$ and $k = 50$ and a threshold corresponding to the 90th percentile of the distribution of distances from a point to its centroid yield boundary points representative of the deficiencies in the model. However, these values are easily adjusted early in the iterative process to accommodate for the specific needs of the mission. Additionally, calculation of boundary points with *k-NN* can be computationally expensive. In this project, the scikit-learning library for Python was used and the computational cost of *k-NN* using a *K-D Tree* approach is *O(DNlog(N))*, where *D* is the number of dimensions and *N* is the number of samples [40]. For that reason, in our experiments, the resulting dense cloud from each iteration was randomly sub-sampled to 20,000 points before being analyzed with *k-NN*.

### 2.4. Planning of Next Best View

After the boundary points in the dense cloud have been identified, the boundary points are classified in clusters using the density-based spatial clustering of applications with noise (DBSCAN) algorithm [41,42]. DBSCAN has four attributes for classifying deficiencies in a point cloud. First, DBSCAN is designed to classify spatial data with minimal input parameters (domain knowledge). Second, DBSCAN is an unsupervised machine learning technique, meaning that DBSCAN does not classify clusters based on previously known shapes but rather can detect clusters of arbitrary shapes. Third, DBSCAN is optimized for large spatial data sets. Finally, DBSCAN classifies the points that cannot meet the density threshold of any cluster as noise. This removes unnecessary boundary points from consideration as camera views. The DBSCAN algorithm has been used for recognition of retinal cells in Spoorthy et al. [43], classification of brain scans in Baselice et al. [44], shoe recognition in Paral et al. [45], and defect detection in semi-conductor wafers in Jin et al. [46].

The core idea of the DBSCAN algorithm, as explained by Ester et al. is that for each point of a cluster, the neighborhood of a given radius has to contain a minimum number of points (i.e., density in the neighborhood has to exceed some threshold) [41]. Additionally, there is a maximum distance between two instances to be considered as in the same cluster. DBSCAN uses two input parameters: *min_samples*, and *epsilon* ($\epsilon$) to describe these thresholds of minimum number of points in a neighborhood and maximum distance between clusters respectively. Clustering boundary points with DBSCAN allows for classification of deficiencies based on their position in the Euclidean space. This classification is used to determine where a new camera is necessary because clusters of deficiencies lie on significantly different planes in the Euclidean space.

Figure 4 shows the number of clusters generated using DBSCAN as a function of $\epsilon$. This figure highlights the non linear behavior of clustering deficiencies with DBSCAN and suggests that $\epsilon$ can be adjusted by means of an optimization subroutine.The optimal value of $\epsilon$ would correspond to the maximum number of clusters, ensuring optimal coverage of deficiencies.

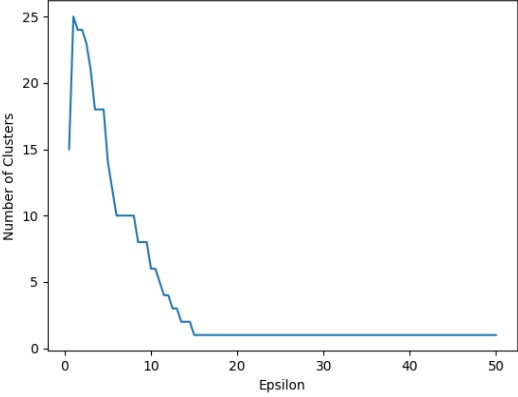

**Figure 4.** Number of clusters in 3D space at different values of $\epsilon$.

### 2.4.1. Optimization with Genetic Algorithm

One approach to finding the optimal value of $\epsilon$ that must be used to identify the maximum number of deficiencies with DBSCAN, is by using gradient free algorithms such as genetic algorithms (GA). Figure 5 shows the clustering that results from using an optimal value of $\epsilon$ as calculated with a GA for a sample point cloud. In this figure, each cluster of deficiencies is identified with an arbitrary color besides brown.

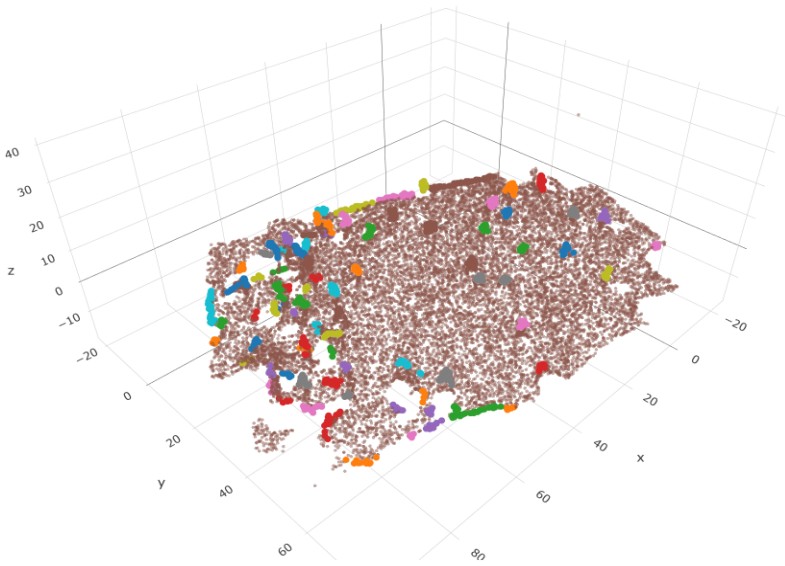

**Figure 5.** Clustering (identified with patches of color) of deficiencies is maximized when optimal values of $\epsilon$ are calculated by genetic algorithms (GA).

### 2.4.2. View and Flight Planning

After the boundary points divide into an optimal number of clusters, it is necessary to calculate a view that will capture these points in a subsequent flight. This is achieved by fitting a plane to all the points in each cluster identified by the DBSCAN algorithm and using the normal to each plane to define the camera orientation. An example of such an approach has been explored by Martin et al. where cameras are planned at each normal of a triangulated mesh, and a set covering problem is solved to eliminate unnecessary cameras [14].

Fitting a plane to a number of points in the Euclidean space is a well understood mathematical problem. In this study, Principal Component Analysis (PCA) fits a plane that fits the points in each cluster [47]. PCA calculates the best fitting plane to a group of points in the Euclidean space by analyzing the variance in the data and projecting the points onto a plane spanned by the two dimensions where the variance is higher. Note that if no clustering is applied to the dense cloud, analysis with PCA renders a single plane that best fits all the boundary points. This singular plane and normal might provide the NBV if the number of photos is minimized. However, in many cases, the area being mapped via SfM is a complex set of faces and the optimal amount of photos should maximize coverage. In the proposed algorithm, DBSCAN approximates as many faces as possible by clustering points of similar density. Therefore, it is desired to maximize the number of clusters at each iteration. The eigenvectors from PCA performed on each cluster calculate the camera orientations or views. The normal vector extends to a safety height with the use of a scaling factor that is tuned manually after the first flight, and elevation and azimuth angles are calculated. Some of the normals calculated result in cameras underground or inside the model or would not reach the specified safety height. These normals are eliminated and no cameras are calculated at those locations. Figure 6 shows a view plan calculated using the methods described above.

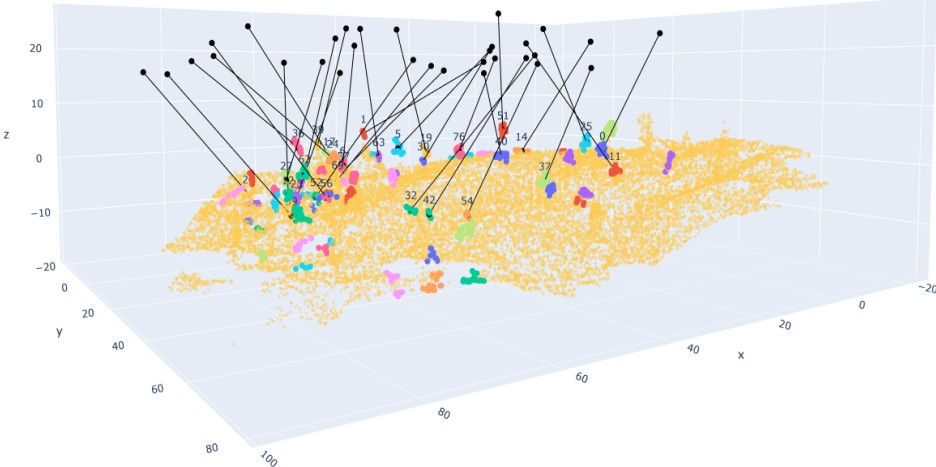

**Figure 6.** Next best view calculated with principal component analysis (PCA) on each cluster.

The calculated views are loaded to the UAV for the next mission, and the photos gathered in this iteration are merged with the previous dense cloud to produce a refined model. The refined model is subsampled to 20,000 points to keep the computational load manageable. This constant point cloud size causes computational load to remain constant for all iterations. The process is repeated until a convergence criterion is met, as detailed in the next section.

### 2.5. Convergence Criteria

The chosen convergence criterion for this algorithm is the orthomosaic resolution of the final model. Figure 7 shows a model converging to 20% improvement from the initial orthomosaic resolution in 5 iterations.

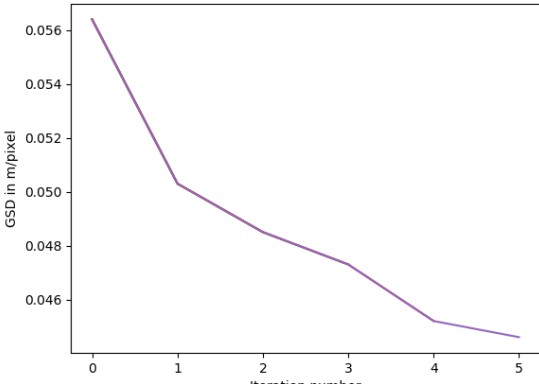

**Figure 7.** Orthomosaic resolution evaluated at each iteration. The resolution improved by 20% from the initial model after 5 iterations, corresponding to approximately 15 photos.

A second convergence criterion could be the asymptotic leveling off of point cloud data points as explored in Section 3.1.1. Leveraging this second convergence criterion is left as a possible avenue for future research.

### 2.6. Simulated and Field Studies

The performance of the proposed iterative modeling strategy was first evaluated using AirSim [48]. AirSim is an open source UAV simulator widely used by many researchers that develop

similar applications and is available on GitHub (https://github.com/microsoft/AirSim) [49,50]. Three different types of environments were used for the simulations (See Figure 8).

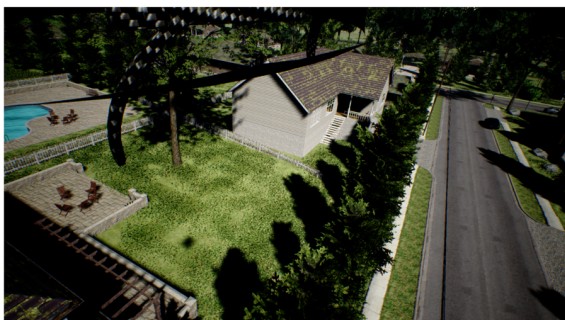

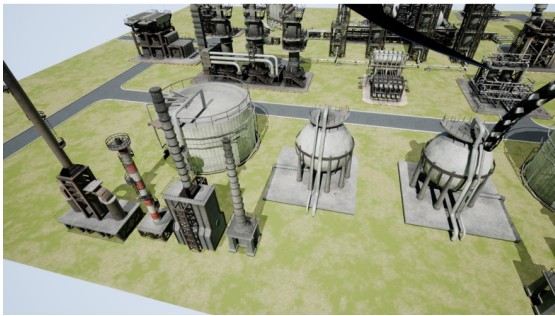
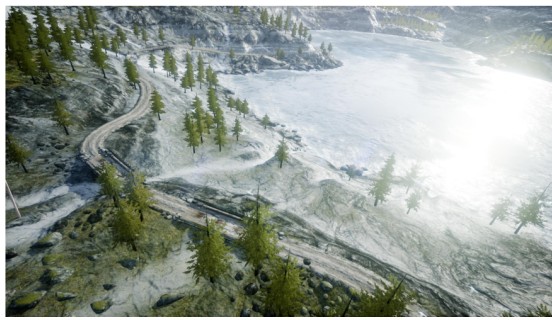

**Figure 8.** AirSim environments: Neighborhood (Top), Refinery (Left), Mountains (Right).

Additionally, the algorithm was used to map an actual municipal water tank in Provo, UT. The tank was selected because it represents typical industrial equipment found in many industrial plants. The water tank is decommissioned, and is located in an area where flying a small UAV does not represent a threat to the integrity of the site or the community. Authorization to fly UAVs for this project was provided by Provo Parks & Grounds.

*2.7. Selection of Equipment and Modeling Software*

The DJI Phantom 4 Pro platform is selected for this case study. DJI drones have provided reliability and accessibility in research applications in the past [51,52]. Furthermore, DJI provides a stable software development kit (SDK: https://developer.dji.com/mobile-sdk/) widely used by mobile developers, which has allowed our research group to build customized applications in an Android mobile device.

Table 1 shows the specifications of the drone and camera selected for this case study have the following specifications.

**Table 1.** Specifications for the DJI Phantom 4 Pro.

| | |
|---|---|
| Sensor | 1″ CMOS |
| Field of View | 77.7° |
| Image Size (Pixels) | 4864 × 3648 |
| Horizontal Viewing Angle | 62.1° |
| Vertical Viewing Angle | 46.6° |
| 3-Axis Gimbal | −90° to +30° |

The modeling software used for rendering SfM models in this case study is Agisoft Metashape 1.5.5 [53]. Agisoft Metashape (previously, Agisoft Photoscan https://www.agisoft.com/) is widely used by the photogrammetry research community [54,55]. Additionally, CloudCompare (http://www.danielgm.net/cc/release/) strengthens quantitative and qualitative analysis of the generated point clouds. Two different computers were used for these experiments, a desktop computer was used for the simulations using AirSim and a laptop computer was used for the field study. The specifications

for each computer are in Tables 2 and 3. The average time for computing a view plan using the proposed iterative approach is 56 and 51 seconds using the computers specified by Tables 2 and 3 respectively. However, this does not include the computational time required to transfer the images from the internal memory of the UAV to the computer or the time required to build the dense cloud. Remote access to a server with specifications similar to those in Table 3 can result in dense cloud build times of up to 33 min. by the fourth iteration. Such integration is outside the scope of this work but recent advances in data transfer protocols and cloud processing promise to make the proposed approach a real-time solution.

**Table 2.** Specifications for the computer used for field study.

| | |
|---|---|
| Software | Agisoft Metashape Professional |
| Software version | 1.6.2 |
| OS | Windows 64 bit |
| RAM | 15.86 GB |
| CPU | Intel(r) Core (TM) i7-6700HQ CPU @ 2.60 GHz |
| GPU(s) | None |

**Table 3.** Specifications for the computer used in simulations.

| | |
|---|---|
| Software | Agisoft Metashape Professional |
| Software version | 1.6.1 |
| OS | Windows 64 bit |
| RAM | 255.97 GB |
| CPU | Intel(r) Core (TM) CPU E5-2680 @ 2.80 GHz |
| GPU(s) | TITAN RTX |

## 3. Results

### 3.1. Results from Simulated Environment

The mountain landscape, neighborhood, and refinery AirSim environments permit quantitative and qualitative analysis of simulated NBV photogrammetry. The refinery simulation has more errors than the neighborhood, and the neighborhood has more errors than the landscape simulations due to increasingly complex objects that hinder resolution and cloud to cloud comparisons. A line of future work could include analysis of more localized cloud to cloud and resolution comparisons instead of holistic analysis; such an adjusted analysis could potentially mitigate comparison errors when many complex objects are part of a model-set. Cloud to cloud comparisons include: exporting point clouds from Metashape to CloudCompare, cropping the point clouds to remove obvious outlier points, point count, and Iterative Cloud Point (ICP) fine alignment. Overall trends hold true and GSD, cloud to cloud comparisons, and qualitative images compliment the evaluation of NBV photogrammetry.

### 3.1.1. Quantitative Results

Figure 9 gives the GSD and iteration number of each dense point cloud for 9 iterations (0–8). The first iteration adds the most resolution, but each successive iteration adds a larger field of view and provides additional detail where new camera views overlap. Note that the GSD for the simulated refinery trends upward instead of downward. Due to the complexity of the objects contained in the refinery environment, such as pipes and stairs, and the small number of photos in the periphery, the reconstruction may 'fail'. Figure 10 demonstrates that the failures are away from the area of interest, but nonetheless contribute to the increasing GSD. By evaluating the resolution in the region of interest instead of the total reconstruction, progress could likely be better observed. as addressed for the physical flight case study in Section 3.2.1. Despite the upward trend, GSD on the refinery environment appears to settle, after enough iterations, to an average value lower than the initial point cloud.

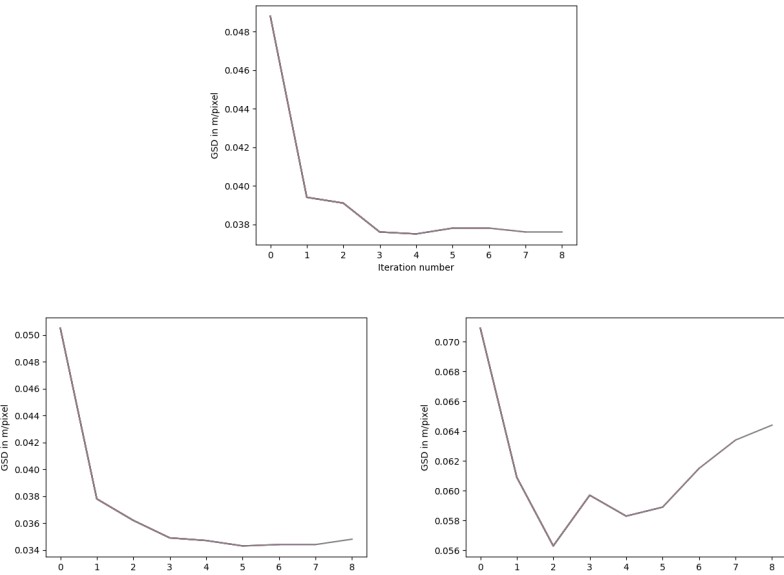

**Figure 9.** Ground sampling distance (GSD) and iteration number: Landscape (**top**), Neighborhood (**left**), Refinery (**right**).

Figure 10 shows the amount of points in each dense cloud after every iteration as well as demonstrates a logarithmic curve fit (approximating zero as 0.1 in the left graph and removing Iteration-0 from the right graph). The $R^2$ value for each regression varies from 0.73 to 0.95 for the landscape, neighborhood, and refinery, and indicates that a logarithmic curve fit naturally matches the simulated data for all three example environments. The increasing number of points is an indication of sufficient overlapping for accurate SfM and of the model increasing both in size and in density. Similarly to GSD, the number of points trend toward a steady value after several iterations, suggesting a limit to the model density that can be obtained with the specified restrictions on safety limits, and camera resolution. In an automated mission, this limiting behavior could serve as a second convergence criterion to prevent the UAV from flying a mission that will not contribute to model improvement.

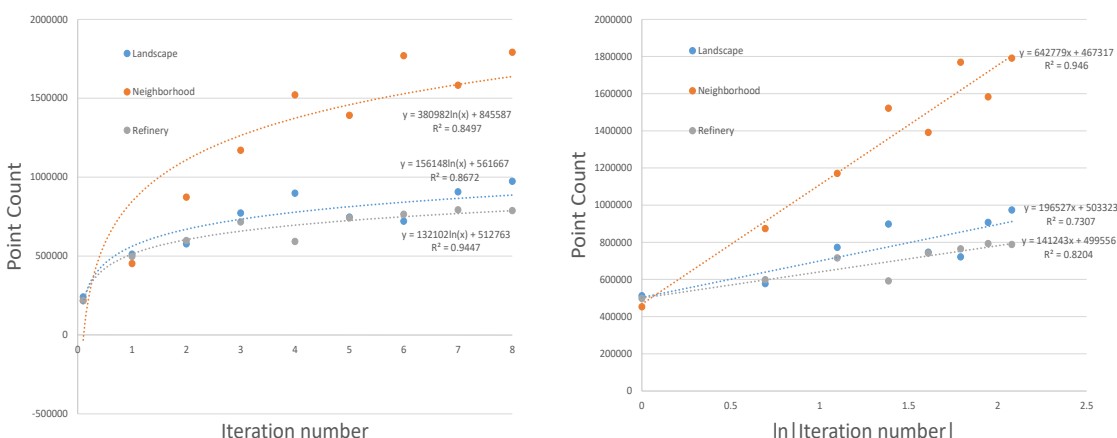

**Figure 10.** Points per iteration and logarithmic regression : Iteration number vs. Point Count (**left**) and ln|Iteration number| vs. Point Count (**right**).

Figure 11 further confirms overlapping, as reported by Metashape after each iteration. In the first few iterations, the area of interest is completely covered by at least 9 photographs that overlap. In the remaining iterations, the photos gathered by the UAV attempt to increase the overall resolution. As mentioned in Section 2: *Methods*, distance between centroid and *k-NN* re-evaluates after each iteration and the 90th percentile of the distribution of distances is selected as the threshold value to

classify a group of points as a deficiency. While collecting more photos inherently contributes to model density, the proposed algorithm seeks to gather photos at targeted areas of lower point cloud density.

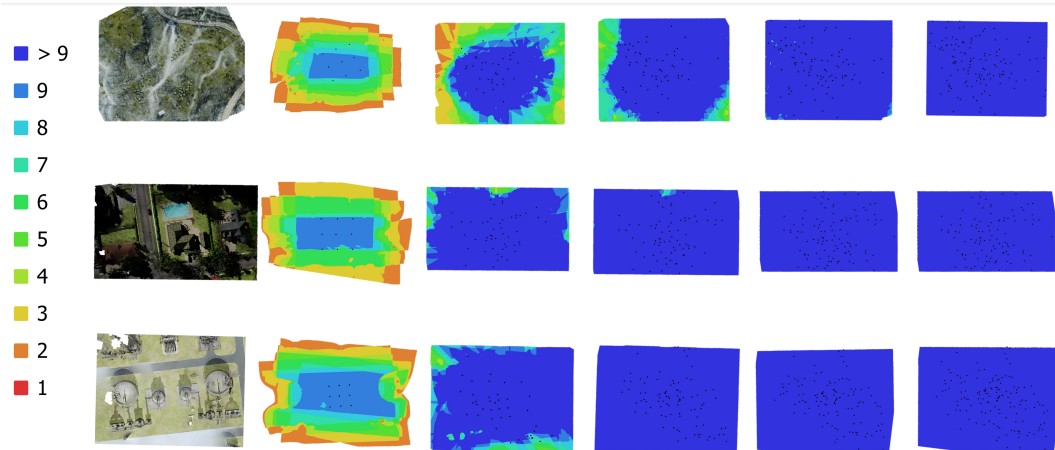

**Figure 11.** Number of images that overlap as reported by Metashape for the first 4 iterations on Landscape (**top**), Neighborhood (**middle**) and Refinery (**bottom**).

### 3.1.2. Qualitative Analysis

Visually, Figures 12 and 13 provide ocular representations of how models evolve with each iteration, showing significant improvement between initial and final models.

Focusing on Figure 12, difficult to recreate objects such as distillation columns and thin towers become recognizable. Model holes and errors visibly close and become more cohesive such as the simulated tank drums, butane spheres, and model edges. Future work could include statistical outlier analysis to remove points that clearly do not form part of the cohesive whole, as removing negative artifacts from point cloud data could improve quantitative comparisons across iterations.

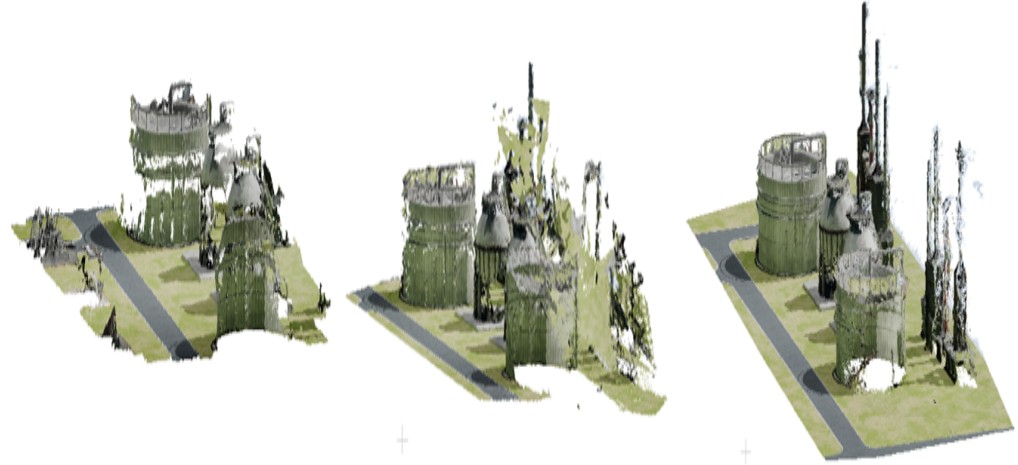

**Figure 12.** Evolution of model in refinery simulated environment; initial Iteration-0 model (**left**), Iteration-4 model (**center**), and final Iteration-8 model (**right**).

Figure 13 of the neighborhood iteration gives qualitative comparisons in CloudCompare (Iterations 0–8 and 7–8). The blue (close), green, yellow, orange, and red (far) dots demonstrate an arbitrary gradient that illustrates overlap between point clouds with the final iteration as the base comparison. The natural coloring of the final iteration (Iteration-8) reveals novel information acquired from additional iterations. Drastic differences between Iteration-0 and Iteration-8 include size and resolution. The differences are not large between Iteration-7 and Iteration-8, but the visual refinement

of model edges and interior holes is apparent throughout the image. Continual iterations increase model size and resolution (on edges and at the interior of the model) to a denser and denser point cloud up to the constraints of hardware capabilities and available time to process data.

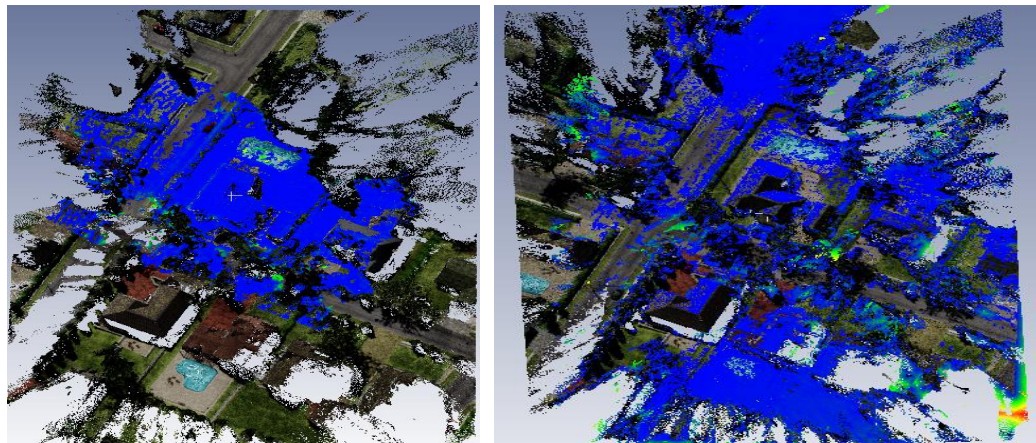

**Figure 13.** Cloud to cloud comparisons of iterations 0–8 (**left**) and 7–8 (**right**) of the AirSim Neighborhood to qualitatively demonstrate photogrammetric progression between iterations. The blue represents similar regions, and demonstrates the similarity between the penultimate and final iterations.

## 3.2. Results from Field Study

As mentioned in Section 2.6, the iterative modelling approach receives validation through a field study of a decommissioned municipal water tank. The initial iteration consists of 3 photos taken manually at a height of 50 *m* and camera orientations determined by an experienced user. The initial set of photos provide the model observed in Figure 14.

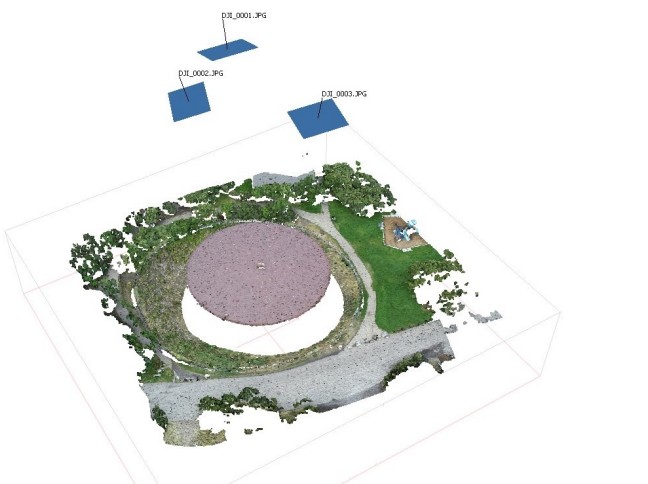

**Figure 14.** Initial model of municipal water tank.

Subsequent iterations are calculated onsite with the equipment specified in Section 2.7. The field study consists of 4 iterations (additional to the initial flight). An example of a generated flight plan appears in Figure 15. Each line corresponds to a camera view. The red lines represent cameras below a predefined safety height and are not included in the mission loaded to the drone.

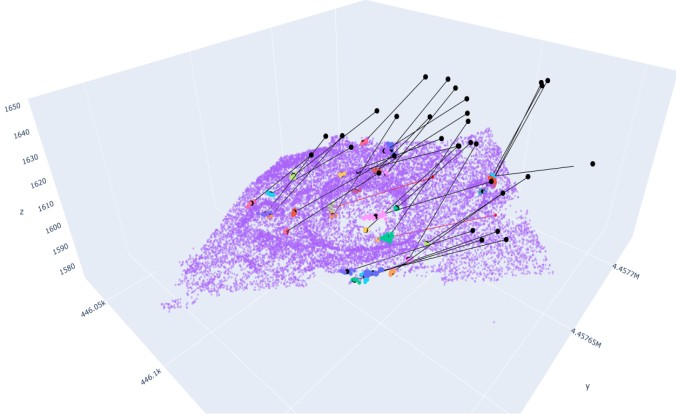

**Figure 15.** Flight plan corresponding to the third iteration.

### 3.2.1. Quantitative Analysis

Figure 16 shows the progression of the GSD with each iteration. The change in GSD after each iteration follows a similar pattern to observations in the simulated refinery environment, as shown in Section 3.1.1. The water tank is similar to petrochemical equipment, and the dense vegetation near the water tank qualitatively matches complex and/or thin objects.

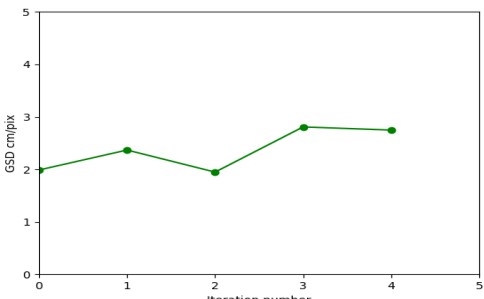

**Figure 16.** GSD in cm/pix of the resulting dense cloud after each iteration (flight) around the water tank.

Similar to analysis provided in Section 3.1.1, Figures 17 and 18 below show the increasing number of points and the overlapping as reported by Metashape, suggesting that the model increases in density with each iteration as fit to a logarithmic regression. Section 3.2.2 shows the dense clouds generated after each iteration, providing visual confirmation that the model evolves around the object of interest.

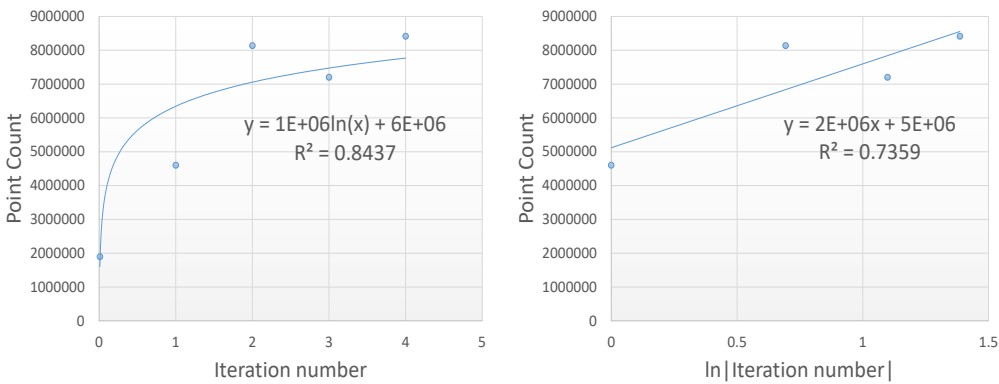

**Figure 17.** Points per iteration and logarithmic regression: Iteration number vs. Point Count (**left**) and ln|Iteration number| vs. Point Count (**right**).

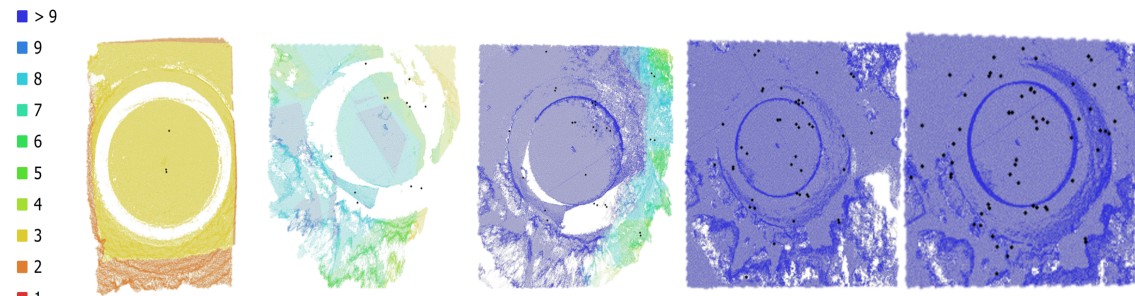

**Figure 18.** Number of images that overlap as reported by Metashape for the 5 iterations around the water tank.

### 3.2.2. Qualitative Analysis

Figure 19 shows the calculated cameras (based on the previous iteration) along with the resulting dense cloud after each iterative flight. The final model, consisting of 105/106 photos successfully aligned, captures the entirety of the water tank.

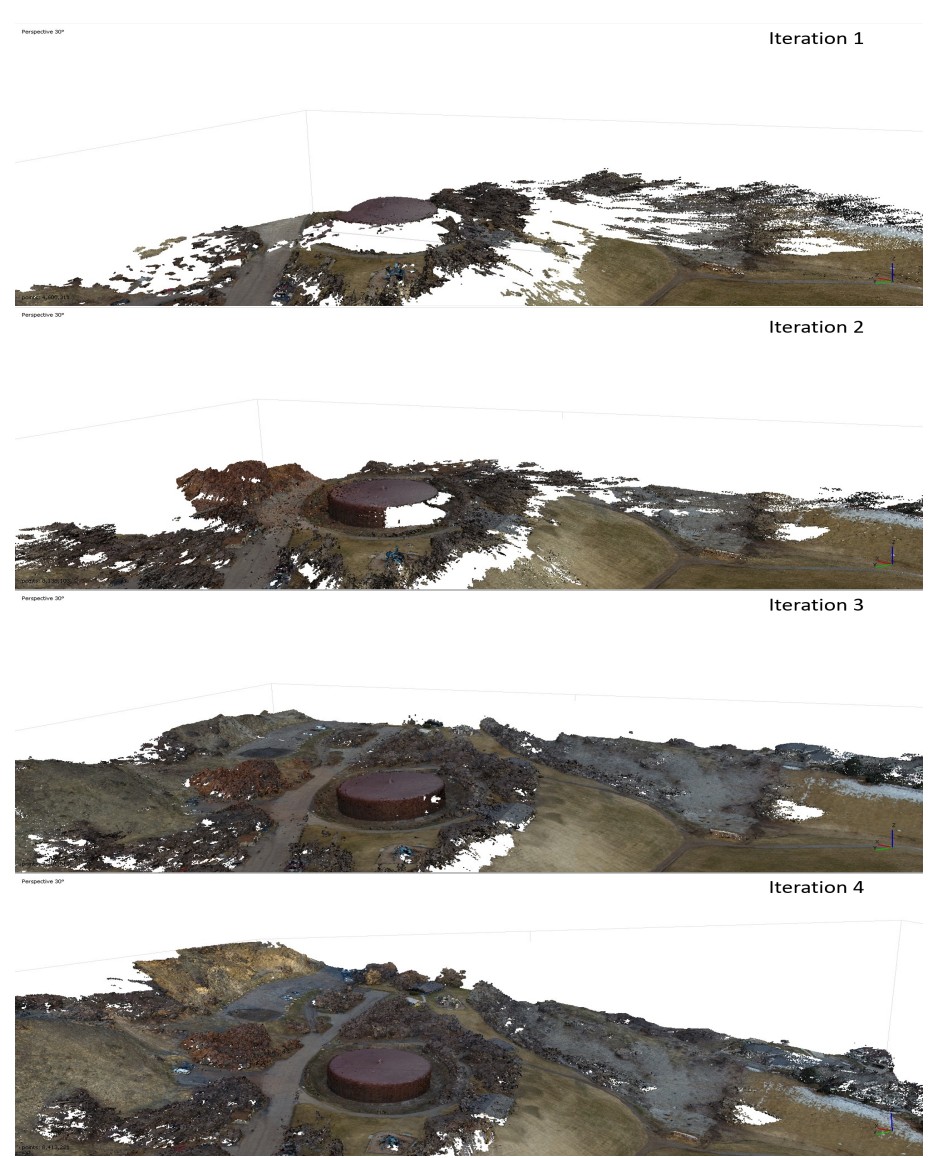

**Figure 19.** Resulting dense cloud after each iterative mission.

## 4. Discussion

The objective of the proposed iterative approach is to automatically map an area without the use of previous models or other *a priori* data). Machine learning methods mimic how a user with domain knowledge would gather images for optimal model reconstruction and decide on a convergence criterion. A discussion on the success and limitations of using the iterative approach follows in subsequent subsections.

### 4.1. Evaluation of Machine Learning Methods

Figure 20 shows a point-to-point comparison of the model generated with the iterative approach and a model generated manually by an experienced user, with Figure 21 explaining the color scale. Figure 22 shows a point-to-point comparison of the model generated with the iterative approach and a model generated automatically with a nadir strategy, with Figure 23 explaining the color scale.

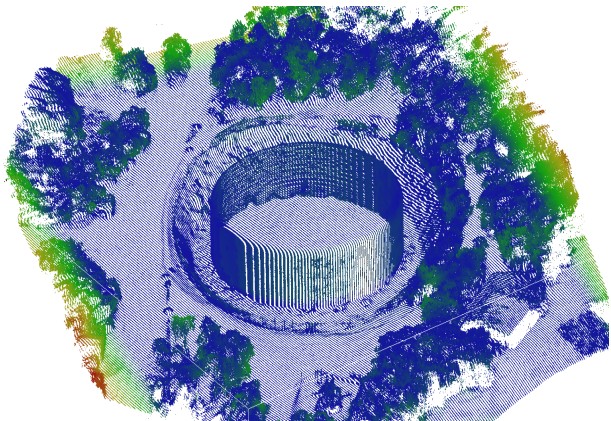

**Figure 20.** Cloud to cloud comparison between the automated iterative mission and a model generated manually by an experienced user.

The manually generated model includes 416 aligned photos and a resulting dense cloud of $93,200,194$ points. The iteratively generated model includes 105 aligned photos and $33,571,630$ points. The distribution of distances between each point is shown in Figure 21, and the distribution further confirms that the object of interest has been successfully and automatically mapped to a level of resolution comparable to that of an expertly flown manual mission. However, the iterative mission uses 63% fewer photographs. Fewer photographs indicate that the tools adequately and artificially reproduce the behavior of an expert user with domain knowledge.

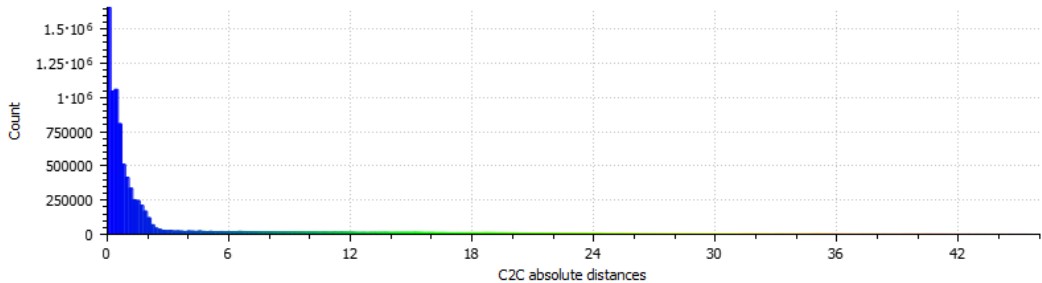

**Figure 21.** Distribution of absolute distances of cloud to cloud analysis of Figure 20.

Figure 21 gives a distribution of absolute distances from CloudCompare for each point of the final iterative model as compared to the high density standard. Although the distances are on a scale internal to CloudCompare, the distribution of points' average distance from the high density standard

indicates that the two models match each other qualitatively because the majority of the points are within two distance units of the high density model.

While the model generated with a nadir flight uses only 63 images, corresponding to 77,511,824 points, significant portions of the model were not captured. The colored portions of Figure 22 highlight areas of the tank left uncaptured by a nadir grid approach.

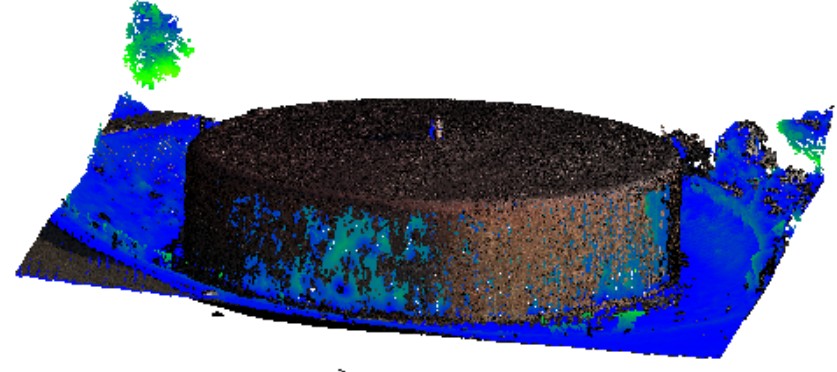

**Figure 22.** Cloud to cloud comparison between the automated iterative mission and a model generated automatically using a nadir flight. Dark colors represent low displacement while warmer colors represent higher displacement.

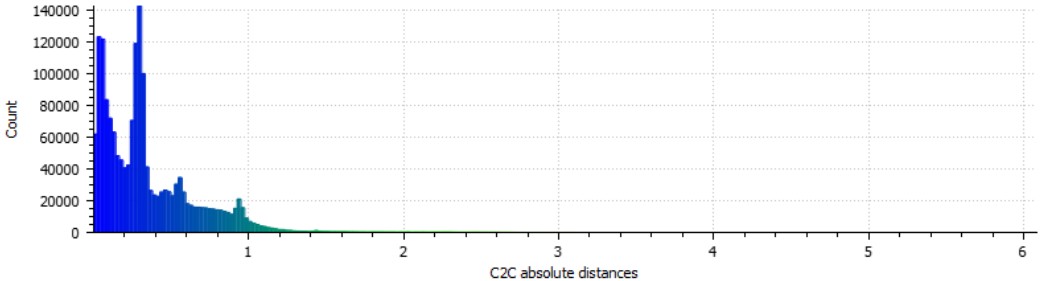

**Figure 23.** Distribution of absolute distances of cloud to cloud analysis of Figure 22.

The comparison between models as discussed above, places the iterative approach as a balanced solution to automatically map an unknown area *Ad summum*, this approach requires minimal domain knowledge to ensure proper coverage and addresses the difficulties of modeling complex structures.

Convergence Criteria in Field Study

While the convergence criterion was essential to the experiments in simulated environments, the field study was manually stopped after the fourth iteration. After this point, the authors determined that the object of interest had been mapped to a satisfactory resolution and that adding more photographs to the model would make the mission impractical, as additional iterations would demand substantially more time to render the models onsite.

Finally, the experiments performed in AirSim prove the concept of autonomous missions, with the drone simulator and the modelling software sharing the same CPU. The same level of interaction between hardware and software in the field would require extensive work outside the scope of this study. However, advancements in technologies such as cloud hosted services and internet bandwidth (capable of handling large amounts of data over a wireless connection), promise to pragmatically develop this technology in future field studies.

## 4.2. Convergence Criteria in Complex Environments

The change in GSD in the refinery environment follows a different pattern than the landscape and neighborhood environments. This observation suggests that using the average orthomosaic GSD as a convergence criterion in a complex environment might be misleading. Images captured at the centroid of the identified deficiencies also capture additional features, and the 3D modelling software attempts to render a reconstruction. However, due to the complexity of the objects in the refinery environment (i.e., piping, vertical structures, stairs) and the limited amount of photographs in the early iterations, the reconstruction fails. Figure 24 shows the progress of a 3D reconstruction in the refinery environment along with its average orthomosaic GSD. Note that on iteration 2 the GSD value goes back up. Figure 25 shows areas of the model that failed to be reconstructed during iteration 2.

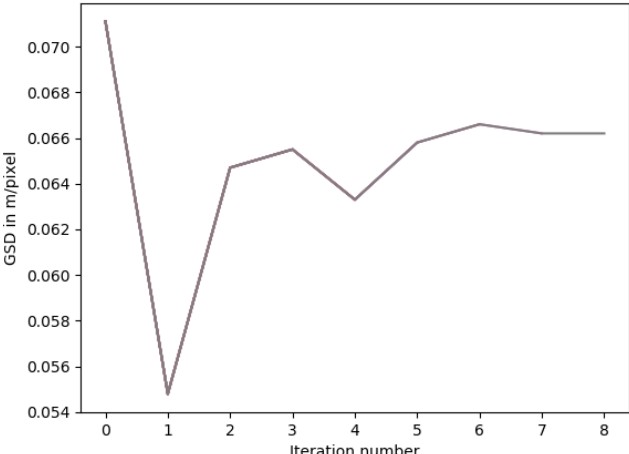

**Figure 24.** Orthomosaic resolution evaluated at each iteration in the refinery environment.

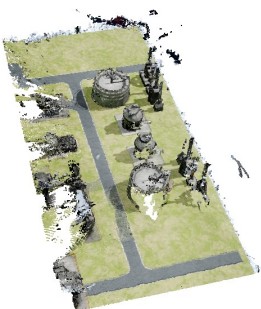

**Figure 25.** Iteration 2 failed to capture additional features, increasing the average GSD.

While failed point clusters in the model are not part of the region of interest, and the region of interest is visually progressing towards a more refined model, the average GSD adversely rises due to the new deficiencies. Therefore, it is proposed to use an alternative metric for convergence when the iterative algorithm is used in complex environments (e.g., petrochemical or manufacturing facilities). The proposed metric consists of trimming the point cloud to the region of interest, and evaluating the local resolution using Equation (1) as proposed proposed in Mineo et al. [38]:

$$\beta = \mu + 2\sigma, \tag{1}$$

where $\beta$ is the local point cloud resolution, $\mu$ is the mean value of the minimum distances between a point and its nearest neighbors, and $\sigma$ is the standard deviation. Figure 26 confirms that evaluation of a localized trimmed-down portion of a point cloud increases in resolution with each iteration. Figure 27

displays the evolution of the model's GSD over 7 iterations. Using this metric, the iterative algorithm may be stopped after a desired level of local resolution has been achieved.

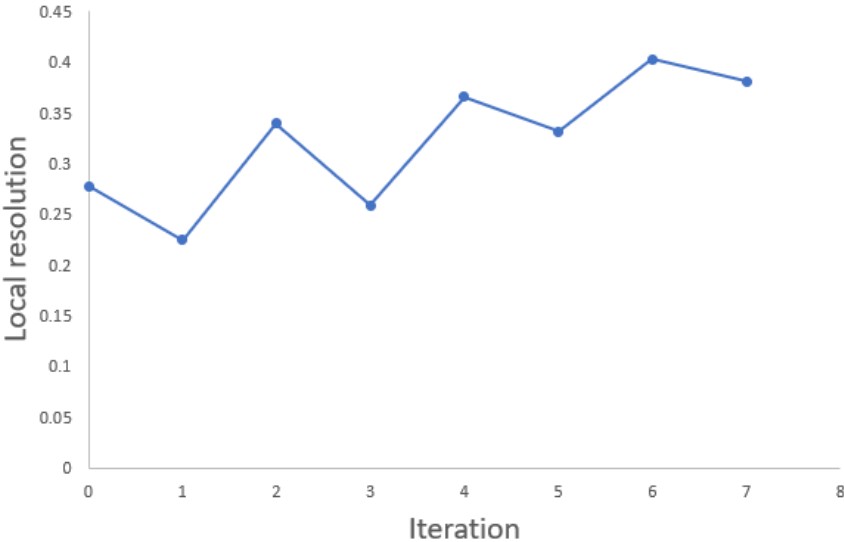

**Figure 26.** Evolution of local point cloud resolution in a refinery environment over 7 iterations.

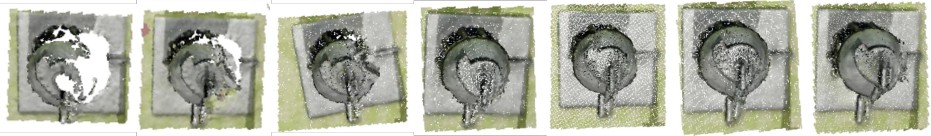

**Figure 27.** Evolution of the area of interest in a 3D model of the refinery environment over 7 iterative flights.

### 4.3. Accuracy of Final Model

The iterative approach highlighted through this study is intended for use in unknown environments where ground control points (GCPs) or ground measurements are not available *a priori*. However, GSD (and potentially the asymptotic behavior of point density), serve to guide the NBV approach to map all existing objects in an area of interest until a resolution limit is reached, suggesting that all objects in the area have been successfully captured. An example application of this strategy would be to use the iterative approach to map an unknown area that is inaccessible to humans due to hazardous conditions but where the GPS coordinates are known. After an initial model has been obtained , it is recommended to use refinement tools, such as the work presented in Okeson et al. That provides targeted reconstruction to desired levels of accuracy [15].

## 5. Conclusions

Iterative modelling can feasibly and autonomously map unknown environments to a desired level of orthomosaic resolution. The performance of the proposed algorithm for iterative modelling is validated in simulated environments (to 3.4 cm per pixel) and in a field study (to less than 3 cm per pixel). The results from the field study of a retired municipal water tank in Rock Canyon Park in Provo, UT show that the iterative approach successfully maps an unknown area within just a few iterations, and the automated approach uses 63% fewer photos than a comparable high quality model (from a manually piloted flight). The field study reveals that compared to models generated with a nadir flight (and some piloted flights), the iterative approach provides better coverage of the area of interest when vertical structures are present.

*5.1. Future Work*

The proposed algorithm consists of different optimization and machine learning subroutines with respective adjustable parameters. The parameter $\epsilon$ used in the DBSCAN clustering algorithm is optimized by the use of GA to maximize the number of deficiencies identified, maximizing coverage in the next iteration. However, optimal parameters for the *k-NN* subroutine are yet to be determined and are left as possible future work. These parameters may also depend on the density of the sampled point cloud. Due to the subsampling procedure, the full point cloud is not observed. Future work may include analysis of the full point cloud and not just a subsampled region. Conversely, analysis of rigorously localized portions of the point cloud could be another avenue for future comparisons. Additionally, benchmarking different optimization subroutines against the GA used in this study, is recommended as future work. A grid independence study or sensitivity analysis of the DBSCAN parameters could yield interesting results. Finally, convergence criteria need to be further refined and explored for complex structures in particular.

This paper is a proof of concept for future in-flight path planning based on real-time model rendering and identification of deficiencies. Once hardware enables access to both photos and computation, this framework will enable in-flight path planning.

**Author Contributions:** S.A. conducted the literature review, conceived the experimental design, developed the methodology, expanded the simulation environment, coded algorithms needed for additional automation, analyzed the data, and oversaw work on the entire study in the lab and in the field. C.A.V. and J.H. vetted the literature review, methodology, algorithms, and analysis. V.N. and J.J. assisted the literature review, running the simulations, and proofreading efforts. K.W.F. brought interdisciplinary geotechnical and UAV expertise from the Research in Optimized Aerial Modeling (ROAM) research group and oversaw the physical UAV missions. J.D.H. provided expert machine learning and automation guidance and leads the Process Research and Intelligent Systems Modeling (PRISM) research group. All authors have read and agreed to the published version of the manuscript.

**Funding:** This work has been funded by the Center for Unmanned Aircraft Systems (C-UAS), a National Science Foundation Industry/University Cooperative Research Center (I/UCRC) under NSF Award No. IIP-1650547 along with significant contributions from C-UAS industry members.

**Acknowledgments:** The authors would like to acknowledge the assistance of Adam Foulk, Bryce Berret and Nicole Hastings for their help with the field study. The authors would also like to acknowledge the assistance of Mark Redd for coding the Genetic Algorithm used in this study.

**Conflicts of Interest:** The authors declare no conflict of interest. The funders provided feedback on industrial relevance of the study and directed the research focus towards multi-scale monitoring. The funders were not involved in the study design or analysis.

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
