# Peer review of "Automated 3D Reconstruction Using Optimized View-Planning Algorithms for Iterative Development of Structure-from-Motion Models"

_remotesensing, doi:10.3390/rs12132169_

Round 1

Reviewer 1 Report

The authors introduce an interesting iterative modeling strategy for additional autonomy in photogrammetry missions. The photos from each iteration combine with previous models to render a more complete point cloud of the scene. The analysis repeats until the model achieves a predefined resolution. This extends the capabilities of Structure for Motion algorithms and provides a framework suitable for robotic autonomous photogrammetry. I think the work in publishable in the current form after addressing the following minor comments:

1) It should be clarified how the initial set of frames is chosen. Are the photos randomly collected in the domain of interest?

2) In section 2.4.2, it is not clear how many points are considered to compute the normals how a safe distance along the normals is calculated to define the camera view points.

3) Table 2 and Table 3 present the details of the computer used for this work. However no indication of the computation time is given. The reader is not informed about the performance of the proposed approach in terms of speed. How does the computation time relate to the iteration number?. Is the approach real-time?.. If not, how far is it for being real-time and what is needed (as future work) to speed it up?

Author Response

Reviewers’ comments to the Authors: 

Reviewer 1 

  1. It should be clarified how the initial set of frames is chosen. Are the photos randomly collected in the domain of interest?

Author Response: We have clarified the method for initial photo selection including the area of interest with 85% overlap. In section 2.2, we have added the sentence: “The initial photo set is collected using a grid pattern that will cover the area of interest with 85% overlap” 

  1. In section 2.4.2, it is not clear how many points are considered to compute the normalshow a safe distance along the normals is calculated to define the camera view points. 

Author Response: Thank you for the note.  All the points in each cluster are considered to fit a plane (resulting in n planes for n number of clusters). The safety height is defined before the mission and a multiplying factor is manually tuned after the first flight. We have added language in the first paragraph of section 2.4.2 to clarify the methods of calculating safe camera positions. 

  1. Table 2 and Table 3 present the details of the computer used for this work. Howeverno indication of the computation time is given. The reader is not informed about the performance of the proposed approach in terms of speed. How does the computation time relate to the iteration number?. Is the approach real-time?.. If not, how far is it for being real-time and what is needed (as future work) to speed it up? 

Author Response: An explanation of computational time has been added to sections 2.3, 2.4.2, and 2.7.  The computational time required for view planning remains constant because the cloud is always subsampled to 20,000 points. The computational times associated with data transfer (from UAV to CPU) and with building the dense cloud depend on the platform used, even with powerful computers, the algorithms used by commercial software to build a dense cloud must be improved to make the approach real time. The revisions follow for clarity: 

Section 2.3, 2.4.2: mention is subsampling point cloud for computational load. " The refined model is subsampled to 20,000 points to keep computational load manageable. This constant point cloud size causes computational load to remain constant for all iterations." 

end of Section 2.7: "The average time for computing a view plan using the proposed iterative approach is 56 and 51 seconds using the computers specified by Tables 2 and 2 respectively. However, this does not include the computational time required to transfer the images from the internal memory of the UAV to the computer or the time required to build the dense cloud. Remote access to a server with specifications similar to those in 3 can result in dense cloud build times of up to 33 min. by the fourth iteration. Such integration is outside the scope of this work but recent advances in data transfer protocols and cloud processing promise to make the proposed approach a real time solution." 

Reviewer 2 Report

The paper presents a method for automatic mission planning of UAV when the scene of interest is not known a priori based on unsupervised machine learning algorithms. A coarse to fine model of the scene is constructed through newly added images (next-best view – NBV approach) until the desired GSD is reached. The method is tested on simulated environments as well as a real case scenario.
The work is undoubtedly of interest, however a crucial issue is completely missing: the only leading parameter to guide the NBV approach is the GSD on the model, while the accuracy requirement of the final reconstruction is not taken into account. The authors are invited to address this important topic. Moreover, a comparative analysis with alternative methods or software to prove the benefits of the proposed approach is advised.
The figures should be improved. Some of them might are not very useful or meaningful (e.g., Fig 4, 5, 22, 24). Others should be better explained (Fig. 6) and improved (Fig 14, 21, 23); for example, metric values of distances should be provided.
It is highly recommended to avoid software related nomenclature (e.g., high quality, line 315 page 22).
The authors may want to better explain why the GSD increases with the iterations (Fig. 10, especially Refinery).

Author Response

Reviewer’s comments:

Reviewer 2 

  1. The work is undoubtedly of interest, however a crucial issue is completely missing: the only leading parameter to guide the NBV approach is the GSD on the model, while the accuracy requirement of the final reconstruction is not taken into account. The authors are invited to address this important topic. Moreover, a comparative analysis with alternative methods or software to prove the benefits of the proposed approach is advised.

Author Response: Thank you! The iterative approach presented in this paper is intended for use in unknown environments where a ground control point or a measurement are not available a priori. However, GSD (and potentially the asymptotic behavior of point density), serve to guide the NBV to map all existing objects in the area of interest until a resolution limit is reached, suggesting that all objects in the area have been successfully captured. An example application of this strategy would be to use the iterative approach to map an unknown area that is inaccessible to humans due to hazardous conditions but where the GPS coordinates are known. 

The novelty of the iterative approach relies on how well the algorithm can approximate what an experienced user would do to complete a deficient model. We have added a paragraph addressing the limitation of the algorithm in regard to final model accuracy and suggesting other tools for accuracy improvement after an initial model has been created. 

  1. The figures should be improved. Some of them might are not very useful or meaningful (e.g., Fig 4, 5, 22, 24). Others should be better explained (Fig. 6) and improved (Fig 14, 21, 23); for example, metric values of distances should be provided.

Author Response: We have revised our figures as follows: 

(fig 4) Removed. Thank you for the suggestion. 

(fig 5) We acknowledge that this figure seems abstract and have added additional description. With the additional context, we feel this figure highlights the need for a nonlinear optimization subroutine. 

(fig 6) We have explained what the color represents within the figure to add clarity 

(fig 14) We have added language to clarify the figure 

(fig 21) We are unsure how to remove the artifacts that occur when downsizing a cloud compare representation of the model. We have added a sentence explaining this and apologize. 

(fig 22) We feel that the qualitative and quantitative information presented in this figure is valuable to the reader. In addition, the data is compared easily with figure 24 which assists comparison of these contrasting data sets. 

(fig 23) We have added information to assist the reader. 

(fig 24) See figure 22. 

  1. It is highly recommended to avoid software related nomenclature (e.g., high quality, line 315 page 22).

Author Response: We have revised our language to follow better practice. The use of High Quality as a software specific nomenclature has been removed.   

  1. The authors may want to better explain why the GSD increases with the iterations (Fig. 10, especially Refinery).

Author Response: Thank you for highlighting this valuable section which required further discussion. We have included an additional section now labeled 4.2 which discusses this in detail.  

Reviewer 3 Report

This paper proposes a UAV-based Structure-from-Motion’s (SfM) iterative strategy, without the use of previous models to initialize the UAV mission, and demonstrates automated iterative mapping of different terrains with convergence to a specified orthomosaic resolution. This iterative UAV photogrammetry also runs in various Microsoft AirSim environments: mountain landscape, suburban neighborhood, and petrochemical plant (refinery) for validation. Some corrections and enhancement should be made before publication.

  1. The novelty of the work should be emphasized in the Abstract and Introduction. Limitations of the method need to be provided in 3. Results and Discussion.
  2. Lin 77-82 What is the “sUAV” stands for? Explain it when “sUAV” appears at the first time?
  3. Lin 181-184 Each paragraph consists of only one sentence that seems not proper for article writing. 2.4.1. Optimization with Genetic Algorithm should be extended for better explanation of optimization purpose and process. Same suggestion is given to 3.2. Results from Field Study.
  4. Line 354 “optimized by the use of GA to maximize the faces of the model where a camera can be calculated,” is ambiguous. How to calculate a camera?
  5. In Figure 1. General workflow for proposed iterative modeling strategy, the components in the flow chart should be actions or criteria. Use the most common shapes used in the flowchart, such as that Terminals are represented as circles, an Input/Output box is represented as a parallelogram, a Process box is represented as a rectangle, a Decision box is represented as a diamond. Please revise it.
  6. Figure 17 and 18 should be polished for better reading.
  7. Reference format should follow the direction of paper preparedness. (see https://www.mdpi.com/journal/remotesensing/instructions)
  8. Differentiate “criteria” and “criterion” in the text.
  9. Author Contributions should be addressed. For a research article with several authors, a short paragraph specifying their individual contributions must be provided. 
  10. Many Recent advancements in remote sensing technologies motivate the use of Unmanned Aerial Vehicles (UAVs) in a variety of aerial imaging tasks. More recent papers relevant to UAVs published in Remote Sensing are suggested to be cited, such as
  • Closing the Phenotyping Gap: High Resolution UAV Time Series for Soybean Growth Analysis Provides Objective Data from Field Trials, Remote Sens. 2020, 12(10), 1644; https://doi.org/10.3390/rs12101644-
  • Semantic Segmentation Using Deep Learning with Vegetation Indices for Rice Lodging Identification in Multi-date UAV Visible Images, Remote Sens. 2020, 12(4), 633; https://doi.org/10.3390/rs12040633
  • A Comparative Study of RGB and Multispectral Sensor-Based Cotton Canopy Cover Modelling Using Multi-Temporal UAS Data, Remote Sens. 2019, 11(23), 2757; https://doi.org/10.3390/rs11232757

Author Response

Reviewer’s comments:

Reviewer 3 

  1. The novelty of the work should be emphasized in the Abstract and Introduction. Limitations of the method need to be provided in 3. Results and Discussion.

Author Response: Thank you. We hope to be clearer and more explicit. We have revised the Abstract as well as the discussion and future work sections according to your recommendations 

  1. Lin 77-82 What is the “sUAV” stands for? Explain it when “sUAV” appears at the first time?

Author Response: We have standardized our acronym usage so that only UAV us used. 

  1. Lin 181-184 Each paragraph consists of only one sentence that seems not proper for article writing. 2.4.1. Optimization with Genetic Algorithm should be extended for better explanation of optimization purpose and process. Same suggestion is given to 3.2. Results from Field Study.

Author Response: We have revised sections 2.4.1 and 3.2 to with additional information and standardized paragraph size. 

  1. Line 354 “optimized by the use of GA to maximize the faces of the model where a camera can be calculated,” is ambiguous. How to calculate a camera?

Author Response: Cameras are calculated by taking the normal from a plane (done for many planes), 'counting' how many points are in the plane (treating seen points as 'covered,' heading from the point cloud along the normal the desired safety distance, and then using that point and that along that normal for camera orientation. The GA finds a variety of potential planes and adjusts through several iterations following principles from genetic patterns in biology to find which planes/normals will 'see' more points. The wording has been changed to clarify that DBSCAN maximizes number of deficiencies identified. 

  1. In Figure 1. General workflow for proposed iterative modeling strategy, the components in the flow chart should be actions or criteria. Use the most common shapes used in the flowchart, such as that Terminals are represented as circles, an Input/Outputbox is represented as a parallelogram, a Process box is represented as a rectangle, a Decision box is represented as a diamond. Please revise it. 

Author Response: Thank you for the feedback. We have changed the figure according to recommendation. 

  1. Figure 17 and 18 should be polished for better reading.

Author Response: While figure 17 was unable to be changed due to persistent artifacts present while viewing models in cloud compare, figure 18 has been adjusted to be more readable and clear 

  1. Reference format should follow the direction of paper preparedness. (see https://www.mdpi.com/journal/remotesensing/instructions)

Author Response: Mendeley was used to organize the references and was imported into LaTeX. Additional references from revision edits have been properly added and synced to the bibliography file and the reference text. Every use of \citet{} was replaced by use of *authors' names* coupled with the \cite{} command just prior to punctuation in the LaTeX type setting environment. A few URLs are kept in the text to facilitate understanding and access/repeatability even when a full reference would not be required. 

  1. Differentiate “criteria” and “criterion” in the text.

Author Response: Additional clarification has been added on criteria and criterion. The primary criterion is the orthomosaic resolution of the final model. A second possible criterion of the leveling off of point quantity in later models is left to future work as a possible route to optimize the process. 

  1. Author Contributions should be addressed. For a research article with several authors, a short paragraph specifying their individual contributions must be provided. 

Author Response: The noted section has been added. Thank You for assisting our compliance. 

  1. Many Recent advancements in remote sensing technologies motivate the use of Unmanned Aerial Vehicles (UAVs) in a variety of aerial imaging tasks. More recent papers relevant to UAVs published in Remote Sensing are suggested to be cited, such as…

Author Response: The 3 suggested papers have been added as citations). It is exciting how quickly progress is occurring with UAVs applied to remote sensing. 

Round 2

Reviewer 2 Report

The efforts for improving the manuscript are highly appreciated, but unfortunately some key concerns expressed in the previous review have not been properly addressed.

It is good to mention that other tools can be used/integrated to better tackle the accuracy issue. However, the camera network problem and its design should be covered, taking into account the resolution/accuracy requirements. The authors may want to refer to some papers on the topic, such as, just to name a few:

Grün, A., 1978. Progress in photogrammetric point determination by compensation of systematic errors and detection of gross errors. Nachrichten aus dem Karten- und Vermessungswesen, Reihe II: Uebersetzungen, Heft Nr. 36, Frankfurt a.M.

James, M.R. and Robson, S., 2014. Mitigating systematic error in topographic models derived from UAV and ground‐based image networks. Earth Surface Processes and Landforms, 39(10), pp.1413-1420.

Nocerino, E., Menna, F. and Remondino, F., 2014. Accuracy of typical photogrammetric networks in cultural heritage 3D modeling projects. International Archives of the Photogrammetry, Remote Sensing & Spatial Information Sciences, 45.

The authors may want to add metric scale bars to figures 14, 21, 23 and try to improve the rendering (e.g., adding normals). The quantization error shown in Figure 21 should be corrected. The authors may want to check that the point cloud is properly imported in CloudCompare, i.e. if a geographic or cartographic reference system is used a false origin should be applied to the point coordinates.

Author Response

Comments and Suggestions for Authors: 

  1. The efforts for improving the manuscript are highly appreciated, but unfortunately some key concerns expressed in the previous review have not been properly addressed. It is good to mention that other tools can be used/integrated to better tackle the accuracy issue. However, the camera network problem and its design should be covered, taking into account the resolution/accuracy requirements. The authors may want to refer to some papers on the topic, such as, just to name a few:

Grün, A., 1978. Progress in photogrammetric point determination by compensation of systematic errors and detection of gross errors. Nachrichten aus dem Karten- und Vermessungswesen, Reihe II: Uebersetzungen, Heft Nr. 36, Frankfurt a.M. 

James, M.R. and Robson, S., 2014. Mitigating systematic error in topographic models derived from UAV and ground‐based image networks. Earth Surface Processes and Landforms, 39(10), pp.1413-1420. 

Nocerino, E., Menna, F. and Remondino, F., 2014. Accuracy of typical photogrammetric networks in cultural heritage 3D modeling projects. International Archives of the Photogrammetry, Remote Sensing & Spatial Information Sciences, 45. 

We feel that the camera network problem is outside the scope of our work, but we have added a brief paragraph to the subsection 1.2 entitled “Optimized View Planning” that cites the recommended sources, notes that additional insight and research would be a good avenue of future work, and summarizes why some of the points in the cited sources may not necessarily apply to the photogrammetry carried out in our paper. 

  1. The authors may want to add metric scale bars to figures 14, 21, 23 and try to improve the rendering (e.g., adding normals). The quantization error shown in Figure 21 should be corrected. The authors may want to check that the point cloud is properly imported in CloudCompare, i.e. if a geographic or cartographic reference system is used a false origin should be applied to the point coordinates.

Thank you for the feedback. The text explains that Figure 21 uses the internal units of CloudCompare (that are arbitrary), but this is still an appropriate statistic because the Figure is used as qualitative evidence and is not meant to be quantitative evidence. Figure 23 is similar to Figure 20, and a different viewpoint has replaced the previous version of Figure 20 to avoid the artifacts of a birds-eye view in CloudCompare after recalculating normals and confirming the reference system. We appreciate the suggestion for how to use CloudCompare to better communicate our results. We have adjusted images as possible to improve quality. 

Reviewer 3 Report

  1. Figure numbering should be updated, since Figure 4 has been removed from the text.
  2. “Figure 6. Clustering (identified with patches of color) of deficiencies is maximized when using optimal values of e as are calculated by GA.” should be modified, because this version shows a confusing plot without delivering clear concept. What does the various color represent? Provide explanation about Figure 6 in the text.
  3. Check the plot and caption of “Figure 11. Points per iteration and logarithmic regression : Iteration number vs. Point Count (left) and ln|Iteration number| vs. Point Count (right).” What do the authors mean by “ln|Iteration number|”? Figure 18 should be polished for better reading, such as regression equations.
  4. Reference format should follow the direction of paper preparedness. Check all abbreviated journal names. (see https://www.mdpi.com/journal/remotesensing/instructions).
  5. “criteria” is a plural noun, and “criterion” is a singular noun, such as Line 415.
  6. Text format of 5.1 Future Work should be checked. Line 411 has been truncated.
  7. There are many wording mistakes in the adding text. Please check it.

Author Response

Comments and Suggestions for Authors: 

  1. Figure numbering should be updated, since Figure 4 has been removed from the text.

Figure numbering has been updated and the figure that used to be Figure 4 has been removed from the text. 

  1. “Figure 6. Clustering (identified with patches of color) of deficiencies is maximized when using optimal values of e as are calculated by GA.” should be modified, because this version shows a confusing plot without delivering clear concept. What does the various color represent? Provide explanation about Figure 6 in the text.

Thank you for the feedback. We agree that the figure could be better explained and an explanation has been added to the text preceding the figure. 

  1. Check the plot and caption of “Figure 11. Points per iteration and logarithmic regression: Iteration number vs. Point Count (left) and ln|Iteration number| vs. Point Count (right).” What do the authors mean by “ln|Iteration number|”? 

Sorry for the confusion, “ln” is the symbol that represents the natural log mathematical operation and the “|” in the “|Iteration number|” text are the absolute value mathematical operation because the inside of a logarithm is always an absolute value to ensure that no solutions of complex or imaginary numbers arise. The text has not been adjusted because additional explanation of the mathematical operators clutters the text, and it is already mentioned that it is a logarithmic regression, so we believe that context of our wording and the linearized version right next to the natural log version of the regression sufficiently explain Figures 11 and 17. “Iteration number” is the iteration number, in other words iteration number is the number that represents which successive UAV mission was occurring to get the data above it. ln|Iteration number| admittedly is an odd name for an axis, but it is what is shown from the given axis. 

  1. Figure 18 should be polished for better reading, such as regression equations.

As Possible, we have polished the figures in the paper given the time limit provided to do revisions. 

  1. Reference format should follow the direction of paper preparedness. Check all abbreviated journal names. (see https://www.mdpi.com/journal/remotesensing/instructions).

The reminder is appreciated; however, we are using Mendeley to organize our reference citations and have checked the provided website and believe that we are in accordance with paper preparedness. LaTeX is using the citation style included in the files that format it according to how the journal requires and automatically orders and structures references accordingly. We believe that the abbreviated journal names are correct. Could you provide an example of what we are referencing incorrectly so that we can fix it? We want to do the references properly. 

  1. “criteria” is a plural noun, and “criterion” is a singular noun, such as Line 415.

The typos have been adjusted to be correct in terms of plurality throughout the paper. 

  1. Text format of 5.1 Future Work should be checked.

We have checked our text formatting and typesetting and believe that all is now in order. 

  1. Line 411 has been truncated.

We believe that the formatting error has been fixed, thank you for spotting it! 

  1. There are many wording mistakes in the adding text. Please check it.

We appreciate the help in finding out how to better use our syntax, diction, and grammar to better communicate our research. We have proofread the most current version of our work and made minor adjustments that are highlighted or struck through in the document.